# FUNCTION FEATURE LEARNING OF NEURAL NETWORKS

## ABSTRACT

We present a Function Feature Learning (FFL) method that can measure the similarity of non-convex neural networks. The function feature representation provides crucial insights into the understanding of the relations between different local solutions of identical neural networks. Unlike existing methods that use neuron activation vectors over a given dataset as neural network representation, FFL aligns weights of neural networks and projects them into a common function feature space by introducing a chain alignment rule. We investigate the function feature representation on Multi-Layer Perceptron (MLP), Convolutional Neural Network (CNN), and Recurrent Neural Network (RNN), finding that identical neural networks trained with different random initializations on different learning tasks by the Stochastic Gradient Descent (SGD) algorithm can be projected into different fixed points. This finding demonstrates the strong connection between different local solutions of identical neural networks and the equivalence of projected local solutions. With FFL, we also find that the semantics are often presented in a bottom-up way. Besides, FFL provides more insights into the structure of local solutions. Experiments on CIFAR-100, NameData, and tiny ImageNet datasets validate the effectiveness of the proposed method.

## 1 INTRODUCTION

Neural networks have achieved remarkable empirical success in a wide range of machine learning tasks (LeCun et al., 1989; Krizhevsky et al., 2012; He et al., 2016) by finding a good local solution. How to better understand the characteristics of local solutions of neural networks remains an open problem. Recent evidence shows that identical neural networks trained with different initializations achieve nearly the same classification accuracy. Are these trained models (local solutions) equivalent? (Li et al., 2016) claimed that neural networks converge to apparently distinct solutions in which it is difficult to find one-to-one mappings of neuron units. (Raghu et al., 2017; Morcos et al., 2018; Kornblith et al., 2019) concentrated on comparing representations of neural networks using the intermediate output of neural networks over a given dataset. These studies provide important insights into the understanding of similarity of neurons by probing and aligning the intermediate output (or neuron activation) representation of data points, but they do not focus on how to directly measure the similarity of function feature representations of neural networks using weights of networks.

In this paper, we propose a Function Feature Learning (FFL) method to measure the similarity between different trained neural networks. Instead of using intermediate activation/response values of neural networks over a bunch of data points, FFL directly learns an effective weight feature representation from trained neural networks. To address the problem of random permutated weights (Figure 1), a chain alignment rule is introduced to eliminate permutation variables. The aligned weights are then learned to project into a function feature representation space by classifying different classes of local solutions. The learned function features can be used to describe the characteristics of local solutions. With FFL, one can validate some assumptions about the similarity of local solutions.

Function feature learning is built upon data feature learning. Given a set of data points, data feature learning is to learn a function $f_i$ that can describe the underlying representations to measure data similarity. Similarly, given a set of data representation functions $\{f_i\}$, function feature learning is to learn a function $F$ that can measure the similarity of $\{f_i\}$. Specifically, an identical neural network with different weights forms a family of functions $\{f_i\}$ that could cover different function types (an

identical neural network with different weights can be used as different function types for different learning tasks in practice). Function feature learning attempts to discover characteristics of functions and thus provides an effective metric for function similarity measure. In this paper, we propose to describe the function feature representation by using weights of neural networks instead of network structures because neural networks often share a common set of functional building blocks, e.g., global/local linear units, activation units, and normalization units.

Overall, we make four main contributions as follows.

- We propose a Function Feature Learning (FFL) method to measure the similarity of identical neural networks trained from different initializations. FFL first addresses the random permutation of weights of neural networks by using a chain alignment rule and then projects the aligned weights into a common space. We find that there exist strong relations between different local solutions optimized by the Stochastic Gradient Descent (SGD) algorithm.

- We investigate function feature representations of Multi-Layer Perceptron (MLP), Convolutional Neural Network (CNN), and Recurrent Neural Network (RNN) on the CIFAR-100, NameData, and tiny ImageNet datasets. With the chain alignment rule, the proposed FFL approach achieves consistent high accuracy for three types of neural networks, which shows the effectiveness of FFL and the soundness of the aforementioned finding.

- We investigate the chain based semantics and the results suggest that the semantics are hierarchical. The projection directions of all layers are arranged in order along with the depth of neural networks. In short, the semantics are presented in a bottom-up way.

- We analyze several factors of neural networks and find that 1) adding more layers or changing the ReLU activation function into leaky ReLU has little impact on the structure of local solutions; 2) changing plain networks into residual networks has some impact on local solutions; 3) SGD often converges to a stable structure of local solutions while the Adam optimizer does not.

***Related Work.*** Neural networks are often regarded as black-boxes due to the non-convexity. To better understand these black-boxes, various approaches provide effective tools for visual interpretability of neural networks (Simonyan et al., 2013; Dosovitskiy & Brox, 2016; Zeiler & Fergus, 2014; Zhou et al., 2015; Selvaraju et al., 2016). These approaches utilized gradient of the class scores with respect to input or de-convolution operations to visualize the attention activations at high-level semantics.

Instead of building visual interpretability foundations between input and output, recent research (Raghu et al., 2017; Morcos et al., 2018; Kornblith et al., 2019) focused on representations of neural networks by exploiting intermediate activations/features to describe the similarity of neural networks. For example, SVCCA (Raghu et al., 2017) used singular value decomposition and canonical correlation analysis tools for network representations and similarity comparison of neural networks. After that, a projection weighted CCA approach was developed for better understanding similarity of neural networks. In (Kornblith et al., 2019), a centered kernel alignment method was proposed to measure the relation between data representational similarity matrices. Our approach concentrates on the function/weight feature representation but not intermediate representations of data points, which is greatly different from these works.

## 2 PRELIMINARIES

In this section, we first introduce related notations and then describe the permutation problem of neural networks.

### 2.1 NOTATION

Let a $L$-layer neural network contain a series of stacked units $\{g_l(x; W_l)\}_1^L$ with global/local linear operations, where $x \in R^{n_0}$ is input and $W_l \in R^{n_{l-1} \times n_l}$ denotes the weights of the $l$-th unit. $n_l$ denotes the number of neurons in the $l$-th layer. We formulate the stacked units as $F_L = \sigma(g_L(\sigma(g_{L-1}(...\sigma(g_1(x; W_1))...; W_{L-1})); W_L))$. $F_L$ is a family of functions that share an identical network structure and $\sigma$ is an activation function. $(W_1, W_2, ..., W_{L-1}, W_L)$ determines the

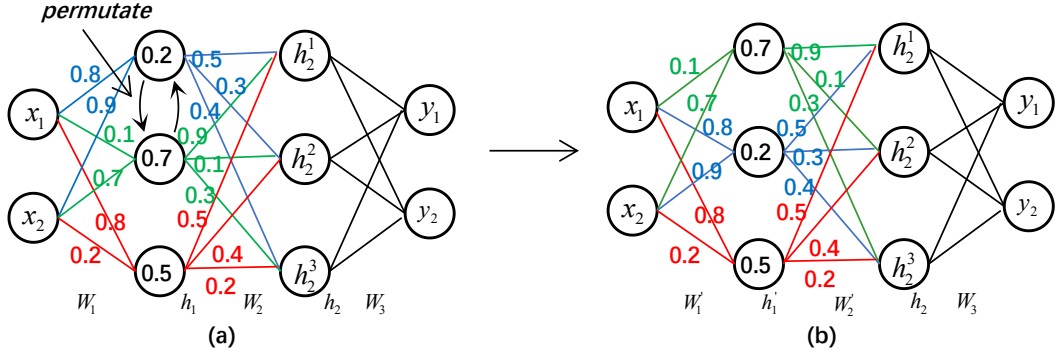

Figure 1: Permutation of neural networks.

function representation of the neural network. We denote $h_l = \sigma(g_l(...\sigma(g_1(x; W_1))...; W_l))$ as the $l$-th hidden vector, $h_l \in R^{n_l}$. When $g_l$ is a local linear/convolutional operation, $F_L$ represents a CN-N. When $g_l$ is a global linear operation, $F_L$ represents an MLP. When $\{g_l(x; W_l)\}_1^L$ share weights, $F_L$ represents an RNN. Because a CNN can be regarded as a patch-based MLP and an RNN can be regarded as a variant version of MLP, we consider MLP for formulation. Besides, the term function, local solution, and trained model will be used synonymously for better understanding.

## 2.2 RANDOM PERMUTATION OF NEURAL NETWORKS

Neurons in the intermediate layers of neural networks (termed intermediate neurons) are often symmetric without any constraint. Neurons in the first and last layers (side neurons) are manually constrained by data and label structures, respectively. Under such constraints, permutating intermediate neurons and their corresponding weights could produce the same output, as mentioned in (Li et al., 2016). We refer to them as neuron permutation and weight permutation.

Permuting neurons and weights according to a certain rule could produce the same output. Because activation functions are often element based operations and do not have any impact on the permutation of neural networks, we consider a 4-layer linear MLP for simplification, as shown in Figure 1 (a). The neural network takes $x = (x_1, x_2)^T$ as input and outputs $y = (y_1, y_2)^T$. The network contains two intermediate hidden layers, denoted as $h_1 = (0.2, 0.7, 0.5)^T$ and $h_2 = (h_2^1, h_2^2, h_2^3)^T$. The weights $W_1$ and $W_2$ are $[0.8, 0.9; 0.1, 0.7; 0.8, 0.2]^T$ and $[0.5, 0.9, 0.5; 0.3, 0.1, 0.4; 0.4, 0.3, 0.2]^T$ (MATLAB-like notation), respectively. Now we permutate the first and second neurons of $h_1$ and permutate the corresponding columns of $W_1$ and rows of $W_2$. We obtain $h_1^{'} = (0.7, 0.2, 0.5)^T$ and $W_1^{'} = [0.1, 0.7; 0.8, 0.9; 0.8, 0.2]^T$ and $W_2^{'} = [0.9, 0.5, 0.5; 0.1, 0.3, 0.4; 0.3, 0.4, 0.2]^T$, as shown in Figure 1 (b). Given any input $x$, the output of these two functions (a) and (b) are equivalent because $W_2^T W_1^T x = (W_2^{'})^T (W_1^{'})^T x$. Neurons and weights are just like flexible nodes and wires that can be easily exchanged. Weight permutation (blue and green wires are exchanged) is exchanged in accord with neuron permutation. Due to neuron permutation and weight permutation, repeating the optimization procedure of identical neural networks would generate different permutations of equivalent local solutions even if we assume neural networks are convex. Therefore, it is important to align the weight permutation of neural networks before analyzing the similarity of local solutions of neural networks.

## 3 METHOD

In this section, we first provide a principle for validation foundation. We then introduce a rule for the alignment of neural networks. Finally, we propose to learn a function feature representation based on the aligned weights.

## 3.1 Learning tells the truth principle

In machine learning, an effective way to learn underlying patterns or rules is to exploit labeled data points to perform supervised learning. However, in some cases, it is difficult to know if a rule is true or an annotation approach is correct. To this, we introduce a *Learning tells the truth* principle. Suppose there exists a learning algorithm such that an assumptive rule learned from a training set can be also well-validated on a test set, then the rule holds. *Learning tells the truth* provides an assumption-learning-validation paradigm to validate some assumptions.

The key point of this principle is to make reasonable assumptions, assign labels to a set of objects and find an effective learning algorithm. In this paper, the assumptions are that local solutions across different runs of an identical neural network converge to a similar local minimum under some conditions (e.g., different optimizers and different neural network structures). Based on the assumptions, an intuitive annotation way is to assign the same label to local solutions of a learning task in different runs. Different learning tasks produce different local solutions and their corresponding labels. We define them as local solution labels or classes. Under such an assumption, we can create a solution set that contains different solution classes by training different task data sets.

We next try to find an effective learning algorithm such that the assumptive rule learned from the training set can be well-validated on the test set. We discuss this step in the next section.

## 3.2 Chain Alignment Rule of Neural Networks

The common approach of aligning weights of neural networks is to transform $(W_1, W_2, ..., W_{L-1}, W_L)$ into a standard form $(W_1^*, W_2^*, ..., W_{L-1}^*, W_L^*)$ that is invariant to weight permutation. However, it is difficult to define such an ideal standard form or directly match two solutions because the structure of local solutions is not only affected by the symmetry of neurons but also determined by the non-convex optimization algorithm. To achieve this, we attempt to eliminate the permutation factors by considering the relations between variables of different layers.

We first consider the weight $W_1$. $W_1$ is a $n_0 \times n_1$ matrix. If we want to permutate neurons of $h_1$ and keep the output unchangeable, we have to permute the columns of $W_1$ and the rows of $W_2$ correspondingly, as illustrated in Figure 1. Given any non-standard $W_1$, suppose there is a column permutation matrix $Q_1 \in R^{n_1 \times n_1}$ such that $W_1$ can be transformed into $W_1^*$. We have

$$W_1 Q_1 = W_1^*. \tag{1}$$

In Eq. 1, we cannot directly solve $W_1^*$, because both $W_1^*$ and $Q_1$ are unknown. Instead, we eliminate the permutation factor $Q_1$ by

$$(W_1 Q_1)(W_1 Q_1)^T = W_1^* W_1^{*T}. \tag{2}$$

Because $Q_1$ is the permutation of the identity matrix $I$ and thus a normalized orthogonal matrix. Hence, $Q_1 Q_1^T = I^{n_1 \times n_1}$. We obtain

$$W_1 W_1^T = W_1^* W_1^{*T}, \tag{3}$$

which is invariant to random permutation $Q_1$.

We then consider $W_2$. $W_2$ could be affected by the column permutation of $W_1^*$ and the row permutation of $W_3^*$. Given any non-standard $W_2$, suppose there are a row permutation matrix $P_2 \in R^{n_1 \times n_1}$ and a column permutation matrix $Q_2 \in R^{n_2 \times n_2}$ such that $W_2$ can be transformed into $W_2^*$. We have

$$P_2 W_2 Q_2 = W_2^*, \tag{4}$$

where $Q_1 = P_2^T$, because the standardization of $W_2$ is jointly affected by $W_1$, as illustrated in Figure 1. $Q_2$ and $P_2$ are orthogonal matrixes. Hence, $Q_1 P_2 = P_2^T P_2 = I^{n_1 \times n_1}, Q_2 Q_2^T = I^{n_2 \times n_2}$. In Eq. 4, it is difficult to eliminate both $P_2$ and $Q_2$. Combining Eqs. 1 and 4, we obtain

$$W_1 Q_1 P_2 W_2 Q_2 = W_1 W_2 Q_2 = W_1^* W_2^* \tag{5}$$

Similar to Eq. 2, we eliminate the permutation factor $Q_2$ by

$$(W_1 W_2 Q_2)(W_1 W_2 Q_2)^T = (W_1^* W_2^*)(W_1^* W_2^*)^T. \tag{6}$$

Hence, we obtain

$$W_1 W_2 W_2^T W_1^T = W_1^* W_2^* W_2^{T*} W_1^{T*}. \tag{7}$$

In this way, we can easily generalize Eq. 7 to the case of the $l$-th layer

$$W_1 W_2 ... W_l W_l^T ... W_2^T W_1^T = W_1^* W_2^* ... W_l^* W_l^{*T} ... W_2^{T*} W_1^{T*}. \tag{8}$$

It is observed that the left of Eq. 8 is independent of permutation factors. We term Eq. 8 as the ***chain alignment rule*** of neural networks. Here, a chain is defined as a sequence of layers of a neural network that begins with the first layer. The $l$-th chain is from the 1-st layer to the $l$-th layer.

### 3.3 FUNCTION FEATURE LEARNING OF NEURAL NETWORKS

Data feature learning is often achieved by minimizing the distance between intra-class data points and maximizing the distance between inter-class data points. Similar to data feature learning, function feature learning can be also achieved by minimizing the distance between intra-class local solutions and maximizing the distance between inter-class local solutions. Here, intra-class local solutions are a family of local solutions trained by similar procedures on the same learning task dataset. Inter-class local solutions are those who are trained on different learning task datasets. For each function (local solution) class, we repeat the training procedure $m_i$ times and thus obtain $M = \sum_{i=1}^{N} m_i$ trained models, where $N$ is the number of solution classes. We then use these local solutions as metadata points to perform function feature learning to measure the function similarity of neural networks.

We investigate the function feature representation based on chains. For the $l$-th chain, the aligned weight $W_1 W_2 ... W_l W_l^T ... W_2^T W_1^T$ with size of $(n_1, n_1)$ is reshaped into a $(n_1 \times n_1, 1)$ vector and then projected into a common function feature space by learning a projection matrix $\Theta_l$. We use the cross-entropy loss for function classification

$$\mathcal{L}_l = -\sum_{i=1}^{N} \mathcal{Y}_i \log(q_i^l) \tag{9}$$

where $\mathcal{Y}_i$ is the $i$-dimensional value of the one-hot label $\mathcal{Y}$. $q_i^l$ represents the probability of the $i$-th function class of the $l$-th chain. We train $L$ function classifiers for $L$ types of chains. Note that we normalize the weights of each layer in MLP during the function feature learning. When measuring the local solution similarity of two neural networks, we extract function feature representation by using projected vectors. We normalize projected vectors and use the cosine similarity to compute the function similarity. We evaluate the chains from $l = 1$ to $L$ and find that the isometric chains of local solutions are strongly related by the SGD optimization algorithm. We empirically evaluate that local solution classification can achieve about 99% top-1 accuracy.

## 4 EXPERIMENT

In this section, we study the effectiveness of the proposed function feature representation of MLP, CNN, and RNN, and validate several assumptions on three datasets, i.e., CIFAR-100 (Krizhevsky & Hinton, 2009), tiny ImageNet (Russakovsky et al., 2015), and NameData (Paszke et al., 2017).

The assumptions in the experiments are that under some conditions the training of the same learning tasks converges to highly similar local solution structures (even though neural networks are non-convex functions) and that of different tasks does not. We use this assumption to label local solutions (weights), which are used as training data points to train a classifier. The learning algorithm is the chain alignment rule plus the linear projection. High test accuracy means that assumptions are reasonable.

Local solution classification is to validate that under the label assumption the rule/knowledge learning from a training set also holds on the test set. Local solution retrieval aims to validate if function feature representation can be used for unseen solution classes, because solution classes between a training set and a test set are non-overlapping in the retrieval setting. These protocols follow image classification and retrieval and thus have the same motivation.

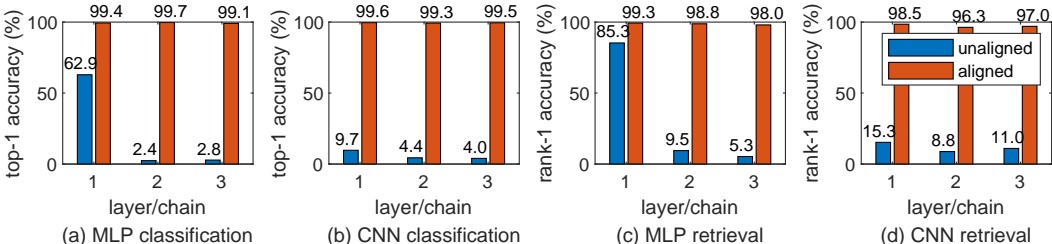

Figure 2: Evaluations on the tiny ImageNet Dataset. "layer" is the x-axis of the unaligned method and "chain" is the x-axis of the aligned method. The following figures are the same.

Specially, Section 4.1, 4.2, and 4.3 assume that under the SGD optimizer condition, local solutions of each task (different runs) are highly similar even though neural networks are non-convex functions. In Section 4.4, the assumptions are slightly different. We change one factor of the baseline to form one new setting each time. Under a new condition, we investigate if the assumption in Section 4.1, 4.2, and 4.3 still holds. We also study if the proposed function feature representation that is trained under the baseline condition is available under another condition. If under both conditions the classification accuracy is high, then the structure of local solutions could nearly not affected by this condition impact factor.

## 4.1 EVALUATIONS ON THE TINY IMAGENET DATASET

The tiny ImageNet dataset, which is drawn from the ImageNet (Russakovsky et al., 2015), has 200 classes, $64 \times 64$ in size. Each class has 500 training images, 50 validation images, and 50 test images. We evaluate the function features of MLP and CNN on the tiny ImageNet dataset. The goal of this experiment is to validate that SGD based local solutions of a learning task are highly similar even though neural networks are non-convex functions. It also aims to demonstrate that the proposed function feature learning is effective.

***Local solution sets.*** We assume that SGD based local solutions of a learning task share the same property and thus have the same label. Based on the assumption, we generate local solution sets. We train a 5-layer convolutional network (PlainNet-5) and a 4-layer MLP (MLP-4) to create two local solution sets for evaluation. The PlainNet-5 network consists of 4 convolutional units and one fully-connected layer. Each convolutional unit contains one convolutional layer with a kernel size of $3 \times 3$, one ReLU function, one BatchNorm layer, and one pooling layer. The MLP-4 consists of 4 fully connected layers (followed by one ReLU functions), among which three layers have 500 hidden neurons and one has $N$ neurons. We split 200 classes into 50 groups as 50 data subsets with solution labels 0~49. Each data subset contains 4 classes. For both MLP and CNN, we repeat the training procedure 100 times to obtain 100 local solutions for each data subset. We generate 5,000 local solutions (weights) for MLP-4 and PlainNet-5, respectively.

***Implementation.*** When generating local solution sets, we use SGD with a batch size of 128. For PlainNet-5, the learning rate starts from 0.1 and is divided by 10 after 30 epochs and we train for 50 epochs. For MLP-4, the learning rate starts from 0.1 and is divided by 10 after 70 epochs and we train for 100 epochs. For saving memory, we resize images into $32 \times 32$ as input and only analyze the first three chains in function feature learning (the fourth chain of CNN takes 72G GPU memory even through the batch size is 1).

When training the function feature representation, we use SGD with a batch size 1. The learning rate is 0.001 and is divided by 10 after 6 epochs. We also set a baseline without weight alignment. For each layer, we directly learn to classify its weights.

***Local solution classification.*** We use local solution classification to validate the similarity of local solutions. We evaluate the performance of local solution classification on MLP and CNN. Similar to image classification, we predict the solutions labels of trained models by using the chain alignment rule and the vector projection. For each solution class, we sample 60 local solutions for training while the other 40 for test. The training set contains 3,000 trained local solutions while the test set contains 2,000 local solutions. We train the function features by classifying 50 solution classes.

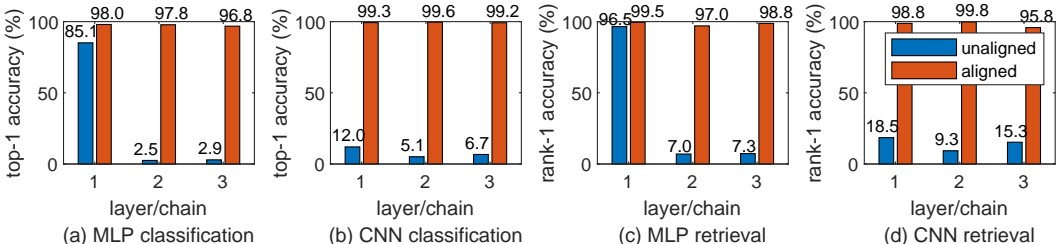

Figure 3: Evaluations on the CIFAR-100 Dataset.

The experimental results are shown in Figure 2 (a) and (b). In local solution classification, our proposed method achieves 99.4%, 99.7% and 99.1% top-1 accuracy using MLP and 99.6%, 99.3% and 99.5% using CNN. The high performance validates the function feature representation of SGD based local solutions with different solutions labels can be closely projected into different fixed points. This suggests SGD based local solutions share similar function features and thus have strong connections. Without using the chain alignment rule, the performance drops significantly, e.g., 62.9%, 2.4% and 2.8% using MLP, 9.7%, 4.4% and 4.0% using CNN. Besides, for the baseline without alignment, the first layer also obtains good performance, because the first layer only has one column permutation variable and thus the row of the matrix contains discriminative information. We also observed that all types of chains can achieve high performance. This suggests that semantics of neural networks are hierarchical in a bottom-up way. In other words, the projection directions of neural networks are arranged in order. Otherwise, the out-of-order projection directions confuse the system and thus lead to poor performance. We finally conclude that local solutions of each learning task or optimization process are highly similar and the chain alignment rule plus linear projection is an effective learning algorithm for validation.

***Local solution retrieval.*** In local solution retrieval, we use the image retrieval metric for local solution retrieval evaluation, i.e., cumulative matching characteristic Gray et al. (2007). Local solution retrieval aims to validate that function feature representation can be generalized to measure the similarity of unseen local solutions. As shown in Figure 2 (c) and (d), our method achieves 99.3%, 98.8% and 98.0% rank-1 accuracy using MLP. For the function feature representation of CNN, the proposed model achieves 98.5%, 96.3%, and 97.0%. Without using the chain alignment rule, the performance drops to 85.3%, 9.5%, and 5.3% on MLP, 15.3%, 8.8%, and 11.0% on CNN. The local solution retrieval results show the robustness of the function representation learning for unseen solution classes. And the high accuracy demonstrates the soundness of the assumption under the SGD optimizer.

## 4.2 EVALUATIONS ON THE CIFAR-100 DATASET

The CIFAR-100 dataset (Krizhevsky & Hinton, 2009), $32 \times 32$ in size, has 100 classes containing 600 images each. There are 500 training images and 100 testing images per class. The assumptions in this section is similar to Section 4.1.

***Local solution sets.*** We train PlainNet-5 and MLP-4 to form two local solution sets on CIFAR-100. The 100 classes of CIFAR-100 is split into 50 groups as 50 data subsets with solution labels 0∼49. Each data subset has 2 image classes. For each data subsets, we repeat the training procedure 100 times to obtain 100 local solutions. Finally, we obtain 5,000 local solutions for MLP and CNN, respectively.

***Implementation.*** The implementation of CIFAR-100 is similar to that of tiny ImageNet. The structure of PlainNet-5 and MLP-4 slightly differs from previous ones because each data subset of CIFAR-100 contains 2 classes and the dimension of the last fully connected layer is 2.

***Local solution classification and retrieval.*** In both local solution classification and retrieval, our method obtains about 98.0% top-1 accuracy and 98.0% rank-1 accuracy, as shown in Figure 3. These results demonstrate the soundness of the assumption and the effectiveness of our proposed function feature learning once again.

Table 1: Evaluations on the NameData Dataset using RNN.

|  | unaligned | aligned |
|---|---|---|
| local solution classification | 17.2% top-1 | 100.0% top-1 |
| local solution retrieval | 21.3% rank-1 | 95.7% rank-1 |

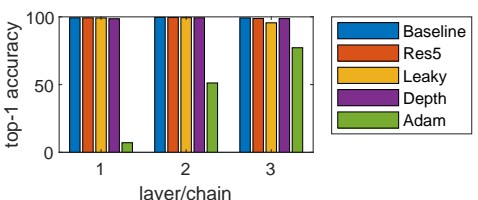
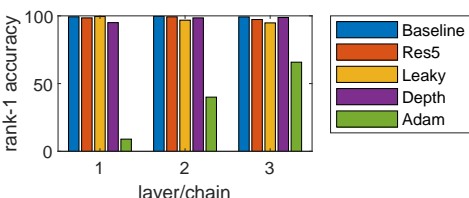

(a) Performance of different settings in classification   (b) Performance of different settings in retrieval

Figure 4: Performance of different settings.

### 4.3 EVALUATIONS ON THE NAMEDATA DATASET

The NameData dataset (Paszke et al., 2017) contains a few thousand surnames from 18 languages of origin. It is used to train a character-level RNN that can predict which language a name is from based on the spelling. The assumption is similar to Section 4.1 but is about Recurrent Neural Network (RNN).

***Local solution sets.*** We train a GRU (Chung et al., 2014) based RNN with two GRU cells (GRU-2) on NameData. A fully connected layer is added after the GRU module for classification. We do not use LSTM (Sundermeyer et al., 2012) because we find that GRU is better than LSTM in our setting. We split 18 classes into 9 groups as 9 data subsets with solution labels 0∼8. Each data subset contains 2 classes. We repeat the training procedure 100 times to obtain 100 local solutions for each class. Finally, we generate 900 local solutions.

***Implementation.*** When generating the local solution set, we use SGD with a mini-batch size of 1. The learning rate starts from 0.1 and is divided by 10 after 7,000 batches and we train for 10,000 batches. When training the function representation, we use SGD with a mini-batch size 1. The learning rate is 0.001 and is divided by 10 after 6 epochs.

***Local solution classification and retrieval.*** We evaluate the function feature representation of RNN with solution classification and retrieval metrics as discussed in Section 4.1. With the chain rule alignment, the proposed method obtains 100.0% top-1 accuracy in the classification setting and 95.7% rank-1 in retrieval setting. Without the alignment approach, the accuracy is 17.2% in classification and 21.3% in retrieval. These results demonstrate the soundness of the assumption and the effectiveness of our proposed function feature learning on RNN.

### 4.4 EFFECT OF DIFFERENT FACTORS

We then study four potential factors that could affect the assumptions made in Section 4.1, 4.2, and 4.3. They are listed as follows. 1) **Baseline**. The baseline is implemented by PlainNet-5 with the ReLU activation function and is optimized by SGD. 2) **Network depth** ("Depth"). Depth is implemented by adding a convolutional unit to PlainNet-5, referred to as PlainNet-6. 3) **Network structure** ("Res5"). Res5 is designed as a 5-layer residual network (ResNet-5). Note that weight size is kept the same. 4) **Optimizer** ("Adam"). Adam is implemented by replacing the SGD optimizer with an Adam optimizer. The initial learning rate is 0.001. 5) **Activation function** ("Leaky"). Leaky is implemented by replacing all of the ReLU activation functions with the leaky ReLU functions. Experiments are conducted on the CIFAR-100 dataset.

We implement four settings by changing one of four factors while keeping the other factors unchanged. For each setting, we train 5,000 local solutions (models) and the solution set is split into the training and test sets as mentioned in 4.2.

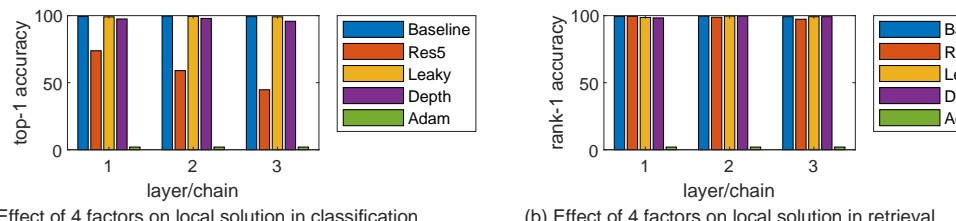

(a) Effect of 4 factors on local solution in classification     (b) Effect of 4 factors on local solution in retrieval

Figure 5: Effect of four factors on the structure of local solution.

***Assumptions and conclusions under four new conditions.*** We investigate if the assumption under the new condition still holds on four new solution sets. In Figure 4 (a) and (b), we have two observations: (1) In both local solution classification and retrieval, using the residual network structure, leaky ReLU activation function or adding one more layer can also obtain high performance of solution classification and retrieval; (2) The Adam optimizer cannot achieve good performance, the reason could be that the Adam optimizer converges to unstable structures of local solutions and thus leads to dissimilar local solutions. Therefore, the assumption in this setting does not hold.

***Effect of four factors on the structure of local solutions.*** Previous discussions focus on the evaluation in a certain setting. In this experiment, we want to find out the effect of these factors on the structure of local solutions. This can be achieved by validating the effectiveness of the function feature representation across different conditions. We train models by using the baseline setting and test by another setting. In Figure 5 (a), we use local solution classification, which aims to validate if the structure of local solutions affected by these factors. The training and test sets share the same learning tasks. We find several key points. First, when applying the function feature representation of PlainNet-5 to that of ResNet-5, the performance drops in the solution classification setting. That means the structure of local solutions of ResNet-5 is changed to some extent compared with PlainNet-5. Second, replacing ReLU by Leaky ReLU or adding one layer to PlainNet-5 nearly does not change the structure of local solutions because the accuracy is still high in solution classification. Third, it is observed that Adam obtains low accuracy. That is, the structure of local solutions of Adam is quite different from that of SGD, the reason could be the unstable adaptive convergence against the convexity of neural networks. In Figure 5 (b), we use local solution retrieval, which aims to validate the effectiveness of function feature learning across conditions. It is observed that the first three factors do not affect the high accuracy of the assumptions, which shows the function features are robust across different conditions to validate these assumptions. Local solutions of Adam optimizer could converge to an unstable structure, which has been discussed. Therefore, function feature learning cannot extract shared features from them.

## 5   CONCLUSION

In this paper, we present a Function Feature Learning (FFL) method that can measure the similarity of non-linear neural networks and thus provides crucial insights into the understanding of the relation between different local solutions of identical neural networks. FFL introduces a novel chain alignment rule for parameter alignment. FFL is used for Multi-Layer Perceptron (MLP), Convolutional Neural Network (CNN), and Recurrent Neural Network (RNN) and evaluated on three datasets. The promising results demonstrate the strong connection between different SGD based local solutions of identical neural networks and the equivalence of projected local solutions by SGD. Besides, the semantics are often presented in a bottom-up way. Finally, FFL provides more insights into the structure of local solutions.

We intend to extend FFL in several directions. First, FFL could be used to measure the transferability between different learning tasks based on the similarity of local solutions. Second, FFL could be used to select diverse learners for ensemble learning based on the dissimilarity of local solutions. Third, FFL could be helpful to find better local solutions in non-convex optimization according to the strong relation between different local solutions.

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
