# OpenReview forum: "Function Feature Learning of Neural Networks"
_ICLR.cc/2020/Conference — Reject_

### Official Review · AnonReviewer1 · 2019-10-23
**Official Blind Review #1**

**Rating:** 3

**Review:**

This paper presents a method called ‘function feature learning’ which do not learn the data distribution but the parameters distribution of several neural networks types. The main idea is to generate many weights from different NNs trained with different random initializations for different subtasks and use them as training data for ‘function feature learning’. The experiments were done on three different datasets.

Overall, the idea is quite interesting and new. However, I’m not 100% sure about the usefulness of the method. The authors claimed to provide more insights of neural networks with their method which I did not see when reading the paper. Furthermore, the authors used a neural networks - a black box model - to provide insights for other neural networks, also black box models. It sounds odd, doesn’t it? Moreover, one assumption from this paper that networks trained with different initializations for the same subtasks produce the same local solution is wrong. Therefore, I’m not 100% sure whether the results produced from all the experiments are trustable.

In sum, I rate this paper as a borderline paper and lean towards rejection due to several aforementioned uncertain points.

**Experience Assessment:**

I have published one or two papers in this area.

**Review Assessment: Checking Correctness Of Derivations And Theory:**

I assessed the sensibility of the derivations and theory.

**Review Assessment: Checking Correctness Of Experiments:**

I assessed the sensibility of the experiments.

**Review Assessment: Thoroughness In Paper Reading:**

I read the paper at least twice and used my best judgement in assessing the paper.

---

> ### Author Response · Authors · 2019-11-10
> **Response to Reviewer #1**
>
> We thank the reviewer for the comments and would like to answer the reviewer’s questions as follows.
>
> Q1: I’m not 100% sure about the usefulness of the method. More insights into neural networks?
> A1: We use the function feature representation together with the “Learning tells the truth” principle to validate some assumptions. We have polished Section 3.1 and Section 4 to make it clear.
>
> In Section 4.1, 4.2 and 4.3, we validate that local solutions of the same learning tasks share a highly similar solution structure even though neural networks are non-convex functions. In Section 4.4, we found 1) local solutions of different network depths (PlainNet-5 and PlainNet-6) to a learning task share a similar structure. 2) Different network structures (plain and residual) partially share similar local solutions to a learning task. 3) Different activation functions (ReLU and LeakyReLU) lead to similar local solutions to a learning task. 4) SGD and Adam optimizers do not share similar local solutions to a learning task.
>
> Due to the non-convexity of the neural network, one could hardly know the properties of local solutions and even do not know if two given solutions are trained from the same learning task. This paper could take one small step towards this goal. Furthermore, one could use the chain aligned and projected solution to validate other useful assumptions.
>
> Besides, we would like to highlight the importance of weight similarity metric that could be under-explored. Weight similarity metrics could provide many potential benefits for machine learning:
> 1)Transfer learning. Transfer learning aims to find similar tasks that contain overlapping knowledge to help the learning of the target task. A good weight similarity metric can tell if two learning tasks are transferable. The low similarity of two tasks leads to negative transfer while high similarity brings positive transfer.
> 2)Ensemble learning. Ensemble learning aims to find a set of diverse learners to further boost performance. A good weight similarity metric can measure the similarity of a set of learners and select diverse ones for ensemble learning.
> 3)Non-convex optimization. A good weight similarity metric could be used to remove redundant local solutions and discover the underlying relations between different local solutions, which provides better gradient descent direction.
>
> We have added these as future works in the conclusion section.
>
>
> Q2: The authors used a neural networks - a black box model - to provide insights for other neural networks, also black box models?
> A2: We analyze the similarity of neural networks by using the chain alignment rule and a linear function (FC layer), but not a black-box model. One can use other traditional linear classifiers instead. We are not sure if we get the right point of the reviewer. If not, please correct us.
>
> Q3: One assumption from this paper that networks trained with different initializations for the same subtasks produce the same local solution is wrong. Therefore, I’m not 100% sure whether the results produced from all the experiments are trustable.
>
> A3: Assumptions could be wrong under some conditions (e.g., ADAM optimizer) while reasonable under some conditions (eg., SGD). We use a “learning tells the truth” principle to validate assumptions. High accuracy means the assumptions are reasonable. To eliminate the reviewer’s concerns, we remove that assumption and use the retrieval setting. We directly use aligned function features (without FC layer for projection, no learning process, normalized)  and cosine similarity to compute the similarity of aligned solutions. The solutions are generated like Section 4.1 and 4.2. We found that without that assumption, the results are still promising. We report these results as follows.
>
> TinyImageNet:
> cmc rank 1,5,10: 97.3,1,1 (using Chain 1)
> cmc rank 1,5,10: 98.8,99.8,1  (Chain 2)
> Cifar100:
> cmc rank 1,5,10: 95.7,99.3,99.8 (Chain 1)
> cmc rank 1,5,10: 97.3,99.6,99.7 (Chain 2)
>
> These show that without the learning process the similarity between aligned local solutions to a learning task is naturally higher than that of different learning tasks, leading to high retrieval results. We only compute the first two chains because weight space is too large (without projecting into the low dimension space).
>
> We would like to share our code and confirm all of the authors’ information is removed. Some related absolute paths could be invalid. The code could take too much time to create a solution set (e.g., 5000 trained models). As a simple example, we suggest the reviewers focus on these files:
> A.Solution set generation:
> --train.py
> --model/mlp.py
> B.Solution classification/retrieval, includes chain alignment rule and linear projection
> -- train_sup.py
> --model/meta_model_mlp.py
>
> The result is easy to reproduce since it is naturally a simple classification/retrieve problem. Anonymous code at: https://anonymous.4open.science/r/74e46ebe-4023-4a85-86b6-19ee20c5070a/

---

### Official Review · AnonReviewer2 · 2019-10-29
**Official Blind Review #2**

**Rating:** 3

**Review:**

I first wanted to thank the authors for their proposed approach in this paper. The paper discusses an interesting idea for quantizing the similarity of neural networks, based on weight similarity. However, overall this assumption is based on the fact that similar layers within a particular architecture will learn similar semantics across different runs (and surprisingly authors add tasks to this as well, which I don't understand why this is the case).

Unfortunately, the paper is hard to read. It is not easy to understand the research questions, and experimental setup. Partly due to overloaded terms, such as “solution” being used for describing multiple different concepts across the paper (solution class, local solution classification, local solution retrieval, none of them particularly well defined in the paper). I highly recommend the authors to describe their findings in a more concrete manner. I did not see any attempts at giving *insights*, mostly numerical comparison.

My concerns with the methodology of the paper are as follows:

1. What are the findings of the paper? Permutation of the neural networks is certainly an area worth studying. However, in this paper, authors make the assumption that weights across the same layer (say layer #2) are somehow always going to learn similar values (except in permutation) across different runs of the model? Major clarification in this area is required.

2. I am not quite sure what the term solution class means, or what the authors want the reviewer to believe it means. Please elaborate. Also solution label? This terminology seems a bit obsolete and cumbersome, unless properly defined at the beginning of the paper.

3. Can you elaborate more on the goals of the experiments? Right now the outcome of the experiments are a bit vague based on lack of hypotheses.

4. Can you elaborate what the goal of local solution classification is? It is not clear if this is simply the classification accuracy of a trained model, or what is vaguely described in section 3.3

I will make a re-evaluation of the paper after the above questions are answered. Overall, I suggest a rewrite of the paper to make claims, experimental hypotheses and design more clear.

**Experience Assessment:**

I have read many papers in this area.

**Review Assessment: Checking Correctness Of Derivations And Theory:**

I carefully checked the derivations and theory.

**Review Assessment: Checking Correctness Of Experiments:**

I carefully checked the experiments.

**Review Assessment: Thoroughness In Paper Reading:**

I read the paper at least twice and used my best judgement in assessing the paper.

---

> ### Author Response · Authors · 2019-11-10
> **Response to Reviewer #2**
>
> We thank the reviewer for the kind review and suggestions. We have revised the paper according to the suggestions and would like to answer the reviewer’s questions as follows.
>
> Q1: What are the findings of the paper? The authors make the assumption that weights across the same layer (say layer #2) are somehow always going to learn similar values?
> A1: We assume that weights across the same chain (say Layer-#1-#2-#3 ) could be affected by the permutation problem in different runs. A chain is defined as a sequence of layers of a neural network that begins with the first layer. Please find the definition in Page 4 of the previous manuscript. That means we compare two nets chain by chain:
> Net1(Layer#1)   <--->  Net2(Layer#1))
> Net1(Layer#1-2)   <--->  Net2(Layer#1-2))
> Net1(Layer#1-2-3)   <--->  Net2(Layer#1-2-3)
> ...
> We also set a baseline that compares two nets layer by layer:
> Net1(Layer#1)   <--->  Net2(Layer#1))
> Net1(Layer#2)   <--->  Net2(Layer#2))
> Net1(Layer#3)   <--->  Net2(Layer#3)
> ...
> Figs. 2, 3 and 4 (Section 4.1, 4.2 and 4.3) show that the chain alignment rule of the proposed method works. Fig. 1 (Section 2.2) shows the motivation of the chain alignment rule.
>
> What are the findings of the paper? In Section 3.1, we introduce a “learning tells the truth” principle. Given a specific assumption, we label a dataset based on the assumption. The dataset is split into a training set and a test set. If a trained model achieves high accuracy on the test set, we say the assumption is reasonable, and vice versa. The findings are listed as follows
> 1)In Section 4.1, 4.2 and 4.3,
> --Assumption: the local solution (weights) of the same learning tasks share a highly similar solution structure even though neural networks are non-convex functions.
> --Setting: based on this assumption, for each task (different runs by SGD), we assign a label to these local solutions. The solution set is split into a training set and a test set.
> --Learning and validation: using the weight alignment and projection can achieve high accuracy (98~99% accuracy) on the test set.
> Conclusion: the assumption holds and the function feature learning works.
>
> 2)In Section 4.4,
> --Assumption:local solutions of different network depths (PlainNet-5 and PlainNet-6) of a learning task share a similar structure. (Learning and validation: yes, high classification/retrieval accuracy)
> --Assumption: different network structures (plain and residual) share similar local solutions of a learning task. (Learning and validation: partially correct, residual nets affect the weights to some extent, moderate classification accuracy)
> --Assumption: different activation functions (ReLU vs. LeakyReLU) lead to similar local solutions of a learning task. (Learning and validation: yes, high accuracy).
> --Assumption: SGD and Adam optimizers lead to similar local solutions of a learning task. (Learning and validation: no, low accuracy).
>
> Except several findings are listed above, some other potential benefits are given in the response to Reviewer #2.
>
> Q2. What the term solution class means, or what the authors want the reviewer to believe it means. Also solution label?
> A2: Solution class or solution label is assigned based on assumptions that local solutions of a learning task (weights in different runs) share the same class while local solutions of different learning tasks have different class labels. Local solutions (trained weights of neural networks) and their assumptive labels can be regarded as data points, which are used for function feature learning. We have provided a clearer definition for solution label and class in Section 3.1. In Section 4.1~4.3 shows the detail for the generation of solution sets.
>
> Q3. Can you elaborate more on the goals of the experiments?
> A3: We have added these in the introduction of Section 4. For each experiment, we have added some words to clarify the goals of the experiments. Specifically, Section 4.1~4.3 assume that under the SGD optimizer condition, local solutions of each task (different runs) are highly similar. In Section4.4, we change one factor of the baseline to form one new setting each time. Under a new condition, we investigate if the assumption in Section 4.1~4.3 still holds. We also study if the proposed function feature representation that is trained under the baseline condition is available under another condition.
>
>
> Q4: Can you elaborate the goal of local solution classification?
> A4:  We have added these in the introduction of Section 4. Local solution classification is to validate that under the label assumption the rule/knowledge learning from a training set also holds on the test set. Local solution retrieval aims to validate if function feature representation can be used for unseen solution classes, because solution classes between a training set and a test set are non-overlapping in the retrieval setting. These protocols follow image classification and retrieval and thus have the same motivation.

---

### Author Response · Authors · 2019-11-10
**Response to all reviewers**

We thank all the reviewers for their helpful comments (while we still have a missing review from R3). We have revised the paper as suggested by the reviewers, and summarize the major changes as follows:
* Clearer explanations about several terminologies required by Reviewer2 are updated in Section 3.1.
* Clearer goals of the experiments and local solution/retrieval required by Reviewer2 are added in Section 4.
* Clearer explanations about the findings/insights of this paper required by Reviewer1 and Reviewer2 are updated in Section 3.1 and Section 4.
* Discussions about the usefulness of the proposed method required by Reviewer1 are added in Section 5.
* Discussions about the soundness of the proposed method required by Reviewer1.
We would like to ask for the reviewers’ suggestions if it is allowed to have one more extra page to include more details and make the paper clearer. We targeted at 8 pages in the initial submission, but according to the reviewers’ comments, it will be helpful to have more details in the main text.

We also try to eliminate each reviewer’s concerns one by one, which can be seen in the corresponding response.

---

### Author Response · Authors · 2019-12-23
**Updated code and Further studies**

Updated code and further studies, please see:

https://github.com/Wanggcong/SolutionSimilarityLearning

This is one of my favorite works! Suggestions are welcomed!

---

### Decision · Program_Chairs · 2019-12-19

**Decision:**

Reject

**Comment:**

This paper tackles an important problem: understanding if different NN solutions are similar or different. In the current form, however, the main motivation for the approach, and what the empirical results tell us, remains unclear. I read the paper after the updates and after reading reviews and author responses, and still had difficulty understanding the goals and outcomes of the experiments (such as what exactly is being reported as test accuracy and what is meant by: "High test accuracy means that assumptions are reasonable."). We highly recommend that the authors revisit the description of the motivation and approach based on comments from reviewers; further explain what is reported as test accuracy in the experiments; and more clearly highlight the insights obtain from the experiments.